# Goal-Reaching Policy Learning from Non-Expert Observations via Effective Subgoal Guidance

**Renming Huang[1], Shaochong Liu[1], Yunqiang Pei[1],**
**Peng Wang[1†], Guoqing Wang[1,3†], Yang Yang[1], Hengtao Shen[1,2]**
[1]School of Computer Science and Engineering,
University of Electronic Science and Technology of China
[2]School of Computer Science and Technology, Tongji University
[3]Donghai Laboratory, Zhoushan, Zhejiang
[†] Corresponding author
hrenming13@gmail.com, p.wang6@hotmail.com, gqwang0420@uestc.edu.cn

**Abstract:** In this work, we address the challenging problem of long-horizon goal-reaching policy learning from non-expert, action-free observation data. Unlike fully labeled expert data, our data is more accessible and avoids the costly process of action labeling. Additionally, compared to online learning, which often involves aimless exploration, our data provides useful guidance for more efficient exploration. To achieve our goal, we propose a novel subgoal guidance learning strategy. The motivation behind this strategy is that long-horizon goals offer limited guidance for efficient exploration and accurate state transition. We develop a diffusion strategy-based high-level policy to generate reasonable subgoals as waypoints, preferring states that more easily lead to the final goal. Additionally, we learn state-goal value functions to encourage efficient subgoal reaching. These two components naturally integrate into the off-policy actor-critic framework, enabling efficient goal attainment through informative exploration. We evaluate our method on complex robotic navigation and manipulation tasks, demonstrating a significant performance advantage over existing methods. Our ablation study further shows that our method is robust to observation data with various corruptions.

**Keywords:** Goal-Reaching, Long-Horizon, Non-Expert Observation Data

## 1 Introduction

Learning goal-reaching policy [1, 2] from sparse rewards holds great promise as it incentivizes agents to achieve a variety of goals, acquire generalizable policies, and obviates the necessity for meticulously crafting reward functions. However, the lack of environmental cues poses significant challenges for policy learning. Online learning methods [3, 4, 5, 6] typically rely on exploring all potentially novel states to cover areas where high rewards may exist, which can lead to inefficient exploration, especially in tasks with long horizons [7]. Alternatively, some methods [8, 9, 10, 11] resort to behavior cloning on external expert data or pre-training with extensive offline data using offline reinforcement learning to obtain an initial policy for online fine-tuning [12]. However, these approaches rely on fully labeled data, which is often expensive or impractical to collect, and faces distribution shift challenges [13] during online fine-tuning.

In this work, we tackle long-horizon goal-reaching policy learning from a novel perspective by leveraging non-expert, action-free observation data. Despite the absence of per-step actions and the lack of guaranteed trajectory quality, this data contains valuable information about states likely to lead to the goal and the connections between them. By extracting such information, we can effectively guide exploration and state transition, achieving higher learning efficiency compared

8th Conference on Robot Learning (CoRL 2024), Munich, Germany.

to pure online learning. Additionally, the accessibility of this data minimizes labeling costs and broadens data sources, rendering our approach practical.

However, this setting poses significant challenges. The absence of action labels precludes the direct learning of low-level per-step policies, while using the final goal as the reward may result in guidance decay, particularly in long-horizon tasks. To address this, we propose the **E**fficient **G**oal-**R**eaching policy learning with **P**rior **O**bservations (EGR-PO) method, which employs a hierarchical approach, extracting reasonable subgoals and exploration guidance from action-free observation data to assist in online learning.

Our method involves learning a diffusion model-based high-level policy to generate reasonable subgoals, acting as waypoints for reaching the final goal. Additionally, we learn another state-goal value function [2] to calculate exploration rewards, thereby encouraging efficient achievement of subgoals and ultimately reaching the final goal. During the pre-training phase, the state-goal value function assists the high-level policy in generating optimal subgoals from non-expert observation data, and the subgoals, acting as nearer goals, addressing issues of guidance decay and prediction inaccuracy [14] encountered by the state-goal value function when dealing with long-horizon final goals. Furthermore, our method seamlessly integrates into the off-policy actor-critic framework [15]. The high-level policy generates subgoals serving as guidance for the low-level policy, while the state-goal value function calculates informative exploration rewards for every transition, fostering effective exploration.

In summary, our contributions are as follows: (1) We offer a fresh perspective on long-horizon goal-reaching policy learning by leveraging non-expert, action-free data, thereby reducing the need for costly data labeling and making our approach practical. (2) We introduce EGR-PO, a hierarchical policy learning strategy comprising subgoal generation and state-goal value function learning. These components work in tandem to facilitate effective and efficient exploration. (3) Through extensive empirical evaluations on challenging robotic navigation and manipulation tasks, we demonstrate the superiority of our method over existing goal-reaching reinforcement learning approaches. Ablation studies further highlight the desirable characteristics of our method, including clearer guidance and robustness to trajectory corruptions.

## 2 Related Work

**Goal-Reaching Policy Learning with Offline Data.** In many real-world reinforcement learning settings, it is straightforward to obtain prior data that can help the agent understand how the world works. Various types of prior data have been widely explored to tackle goal-reaching challenges, including *fully labeled* data, *video* data, *reward-free* data, and *expert demonstrations*. *Fully labeled* data serves as either experience replay [16] during online learning or an offline dataset for pre-training [12, 17, 18, 19, 20, 21], enabling the acquisition of a pre-trained policy. Furthermore, the pre-trained policy can be employed as an external policy to guide exploration [22, 23, 10] or directly fine-tuned online [11, 8]. However, online fine-tuning typically encounters distribution shift [24, 9] and overestimation [8, 11] challenges. On the other hand, *video* data provides a wealth of information that can be directly utilized for learning policy [25, 26, 27], learning state representation for downstream RL [28, 29, 30], guiding the discovery of skills [31], and learning world models for planning and decision-making [32, 33, 34, 35]. Moreover, recent research has showcased the potential of *reward-free* data [36] in accelerating exploration through optimistic reward labeling [6]. In the case of *expert demonstration* data, imitation learning is commonly employed to learn a stable policy [37, 38, 39, 40, 41].

**Online Learning via Exploration.** Sparse rewards hinder learning efficiency, making desired goals challenging to achieve. To address this, boosting exploration abilities allows agents to cover unseen goals and states, facilitating effective learning through experience replay. One standard approach is to add exploration bonuses to encourage exploration to unseen state. Exploration bonuses seek to reward novelty, quantified by various metrics such as density model [42, 43, 44], curiosity [3, 45], model error [46, 47, 5], or even prediction error to a randomly initialized function [4]. Goal-directed exploration involves setting exploratory goals for the policy to pursue. Various goal-selection meth-

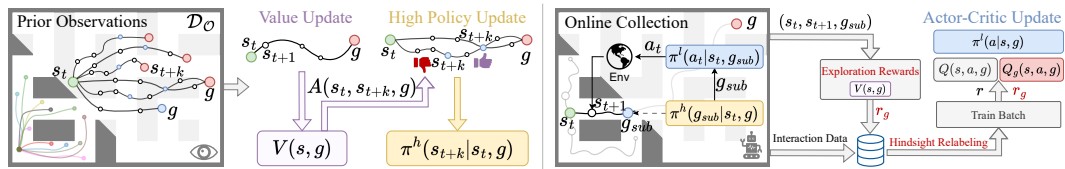

(a) Subgoal Guided Policy Learning from Prior Observations  (b) Efficient Online Learning with Subgoal Guidance

Figure 1: **Overview of EGR-PO**. (a) Our method is composed of two key learning components: a state-goal value function designed for informative exploration and a high-level policy to generate reasonable subgoals. (b) Integrating the two components into the actor-critic method, where the learned state-goal value function provides exploration rewards to encourage meaningful exploration, and the reasonable subgoals provide clear guidance signals.

ods have been proposed, such as frontier-based [48], learning progress [49, 50], goal difficulty [51], "sibling rivalry" [52], value function disagreement [53], go-explore framework [54, 55]. Our method falls under goal-directed exploration, but our reasonable goals are learned from prior observations.

## 3 Preliminaries

**Problem Setting.** We investigate the problem of goal-conditioned reinforcement learning, which is defined by a Markov decision process $\mathcal{M} = (\mathcal{S}, \mathcal{A}, \mu, p, r)$ [56], where $\mathcal{S}$ denotes the state space, $\mathcal{A}$ denotes the action space, $\mu \in P(\mathcal{S})$ denotes an initial state distribution, $p \in \mathcal{S} \times \mathcal{A} \to \mathcal{P}(\mathcal{S})$ denotes a transition dynamics distribution, and $r(s, g)$ denotes a goal-conditioned reward function. We assume that we have an additional observation dataset $\mathcal{D}_{\mathcal{O}}$ that consists of state-only trajectories $\tau_s = (s_0, s_1, \ldots, s_T)$. Our goal is relay on $\mathcal{D}_{\mathcal{O}}$ to help learn an optimal goal-conditioned policy $\pi(a|s, g)$ that maximizes $J(\pi) = \mathbb{E}_{g \sim p(g), \tau \sim p^\pi(\tau)}[\sum_{t=0}^{T} \gamma^t r(s_t, g)]$ with $p^\pi(\tau) = \mu(s_0) \prod_{t=0}^{T-1} \pi(a_t \mid s_t, g)p(s_{t+1} \mid s_t, a_t)$, where $\gamma$ is a discount factor and $p(g)$ is a goal distribution. In our method, the policy is formulated as a hierarchical policy $\pi(a|s, g) = \pi^h(g_{sub}|s, g) \circ \pi^l(a|s, g_{sub})$.

**Implicit Q-learning.** Kostrikov et al. [9] proposed a method called Implicit Q-Learning (IQL) that circumvents the need to query out-of-sample actions by transforming the $\max$ operator in the Bellman optimal equation into expectile regression. Specifically, IQL trains an action-value function $Q(s, a)$ and a state value function $V(s)$ with the following loss:

$$\mathcal{L}_V = \mathbb{E}_{(s,a) \sim \mathcal{D}} \left[ L_2^\tau(\bar{Q}(s, a) - V(s)) \right], \mathcal{L}_Q = \mathbb{E}_{(s,a,s') \sim \mathcal{D}} \left[ (r_{(s,a)} + \gamma V(s') - Q(s, a))^2 \right], \quad (1)$$

where $r_{(s,a)}$ represents the reward function, $\bar{Q}$ represents the target Q network [57], and $L_2^\tau$ denotes the expectile loss with a parameter $\tau$ belonging to the interval $[0.5, 1)$. The expectile loss $L_2^\tau(x)$ is defined as $|\tau - \mathbb{1}(x < 0)|x^2$, which exhibits an asymmetric square loss characteristic by placing greater emphasis on penalizing positive values compared to negative ones.

**Advantage-Weighted Regression (AWR).** AWR [58] considers policy optimization as a maximum likelihood estimation problem within an Expectation-Maximization [59] framework. It utilizes the advantage values to weight the likelihood, thereby encouraging the policy to select actions that lead to large Q values while remaining close to the data collection policy. Given a collection dataset $\mathcal{D}$, the objective of extracting a policy with AWR is formulated as follows [12, 60, 61]:

$$J_\pi(\theta) = \mathbb{E}_{(s,a) \sim \mathcal{D}} \left[ \exp(\beta \cdot A(s, a)) \log \pi_{\theta_\pi}(a|s) \right], \quad (2)$$

where $\beta \in \mathbb{R}_0^+$ denotes an inverse temperature parameter, $A(s, a) = Q(s, a) - V(s)$ represents the extent to which the current action is superior to the average performance.

## 4 Efficient Goal-Reaching with Prior Observations (EGR-PO)

The overview of our method is illustrated in Figure 1. In Section 4.1, we extract guidance components from observations, which involves training a state-goal value function $V(s, g)$ to encourage informative exploration and learning a high-level policy $\pi_\phi^h(g_{sub}^h|s, g)$ that generates reasonable subgoals. In Section 4.2, we illustrate how the learned components collaborate to enhance the learning of the online low-level policy $\pi_\theta^l(a_t|s_t, g_{sub}^h)$. Algorithm 1 presents a sketch of our method, implementation details can be found in the Appendix.

---

**Algorithm 1** Efficient Goal-Reaching with Prior Observations

---

1: **Input:** Observation dataset $\mathcal{D}_\mathcal{O}$
2: **while** not converged **do**
3:  Sample batch $(s, s', g) \sim \mathcal{D}_\mathcal{O}$
4:  Update state-goal value network $V(s, g)$ with Equation (3) # Train state-goal value function
5:  Sample batch $(s, g_{sub}, g) \sim \mathcal{D}_\mathcal{O}$
6:  Update high-level policy $\pi_\phi^h(g_{sub}|s, g)$ with Equation (7) # Extract high-level policy
7: **end while**
8: **for** each environment step **do**
9:  Execute action $a \sim \pi^l(a|s, g_{sub}^h)$ with subgoals $g_{sub}^h \sim \pi^h(g_{sub}^h|s, g)$
10:  Calculate exploration reward $r_g$ with Equation (9), store transition to the replay buffer $\mathcal{D}$
11:  # Actor Critic Style Update
12:  Sample transition mini-batch $\mathcal{B} = \{(s, a, s', r, r_g, g_{sub})\} \sim \mathcal{D}$ with hindsight relabeling
13:  Update $Q$ network and $Q_g$ network by minimize Equation (8) and Equation (10)
14:  Extract online policy $\pi^l$ with Equation (11)
15: **end for**

---

## 4.1 Subgoal Guided Policy Learning from Prior Observations

**Learning state-goal value function for informative exploration.** Learning a value function from prior data encounters overestimation [8] caused by out-of-distribution, we adopt the IQL approach, which prevents querying out-of-distribution "actions". Specifically, we utilize the action-free variant [28, 62] of IQL to learn the state-goal value function, denoted as $V(s, g)$:

$$\mathcal{L}_V = \mathbb{E}_{(s,s',g)\sim\mathcal{D}_\mathcal{O}} \left[ L_2^\tau(r + \gamma \cdot \bar{V}(s', g) - V(s, g)) \right]. \tag{3}$$

The learned state-goal value function is unreliable with increasing distance, as discussed in Section 5.3. Relying solely on it for long-horizon tasks is ineffective and potentially harmful. To address this, we learn another policy that generates nearby and reasonable subgoals, which enhance the accuracy of predicted values and provide clearer guidance signals.

**Learning to generate reasonable subgoals.** We use the diffusion probabilistic model to perform behavior cloning for subgoal generation. Previous studies [63, 64] have demonstrated the robustness of diffusion probabilistic models in policy regression. We represent our diffusion policy using the reverse process of a conditional diffusion model as follows:

$$\pi_\phi^h(g_{sub}^h|s, g) = p_\phi(g_{sub}^{0:N}|s, g) = \mathcal{N}(g_{sub}^N; \mathbf{0}, \mathbf{I}) \prod_{i=1}^{N} p_\phi(g_{sub}^{i-1} \mid g_{sub}^i, s, g). \tag{4}$$

We follow DDPM [65] and train the $\epsilon$-conditional model by optimizing the following objective:

$$J_{BC}(\phi) = \mathbb{E}_{i\sim\mathcal{U},\epsilon\sim\mathcal{N}(\mathbf{0},\mathbf{I}),(s,g_{sub},g)\sim\mathcal{D}_\mathcal{O}} \left[ ||\epsilon - \epsilon_\phi(\sqrt{\bar{\alpha}_i}g_{sub} + \sqrt{1 - \bar{\alpha}_i}\epsilon, s, g, i)||^2 \right], \tag{5}$$

where $\epsilon$ is noise following a Gaussian distribution $\epsilon \sim \mathcal{N}(\mathbf{0}, \mathbf{I})$, $\mathcal{U}$ is a uniform distribution over the discrete set as $\{1, ..., N\}$. Following HIQL [14], we sample goals $g$ from either the future states within the same trajectory or random states in the dataset. Similarly, we sample subgoals $g_{sub}$ from either the future $k$-step states $s_{t+k}$ within the same trajectory.

However, due to the prior data is not from expert, the subgoals generated through behavior cloning are not necessarily optimal. With the state-goal value function learned by Equation 3, we can improve the policy with the following variant of AWR [58]:

$$J_I(\phi) = \mathbb{E}_{(s,g_{sub},g)\sim\mathcal{D}_\mathcal{O}}[\exp(\beta \cdot \tilde{A}(s, g_{sub}, g)) \cdot \log \pi_\phi^h(g_{sub} \mid s, g)], \tag{6}$$

we approximate $\tilde{A}(s, g_{sub}, g)$ as $V_\theta(g_{sub}, g) - V_\theta(s, g)$, which assists extract subgoals with higher advantage without deviating from the data distribution. The final objective function is a linear combination of behavior cloning and policy improvement:

$$\pi_\phi^h = \arg\min_\phi J(\phi) = \arg\min_\phi(J_{BC}(\phi) - \alpha \cdot J_I(\phi)), \tag{7}$$

where $\alpha$ is a hyperparameter used to control the diversity of subgoals.

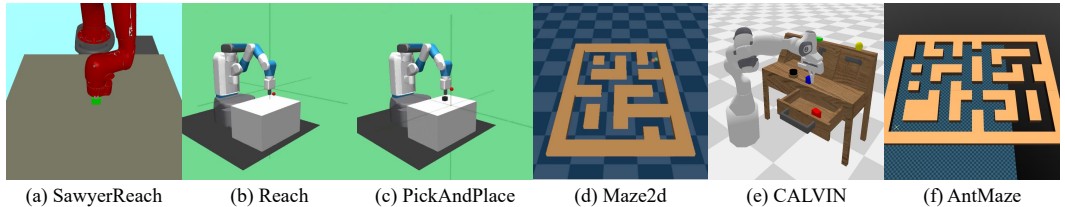

| (a) SawyerReach | (b) Reach | (c) PickAndPlace | (d) Maze2d | (e) CALVIN | (f) AntMaze |

Figure 2: We study the robotic navigation and manipulation tasks with sparse reward.

## 4.2 Efficient Online Learning with Subgoal Guidance

The learned high-level policy and state-goal value function naturally integrate into the off-policy actor-critic paradigm [15, 66, 67, 68, 69]. Actor-critic framework simultaneously learns the action-value function $Q$ [70] and the policy network $\pi_\theta^l$ [71]. The action-value network $Q$ is trained with temporal difference [72]:

$$\mathcal{L}_Q = \mathbb{E}_{(s,a,s',g,r)\sim\mathcal{D},a'\sim\pi_\theta^l(s',g)}\left[\left\|r + \gamma \cdot \bar{Q}(s',a',g) - Q(s,a,g)\right\|^2\right], \tag{8}$$

where $\bar{Q}$ is the target network, $r$ is the environmental reward. We introduce additional exploration rewards $r_g$ to encourage informative exploration. The design of the exploration reward function is based on the learned state-goal value function:

$$R_g(s,s',g) = \tanh(\eta \cdot (V_\theta(s',g) - V_\theta(s,g))), \tag{9}$$

where $\eta$ is a scaling factor. Different from previous methods [4, 73, 74, 75, 76], we do not directly add $r_g$ to the environmental reward $r$. The exploration reward has taken the future into consideration, thus, we introduce an additional guiding Q function $Q_g(s,a,g)$, which directly approximates the exploration reward:

$$\mathcal{L}_{Q_g} = \mathbb{E}_{(s,a,r_g,g_{sub}^h)\sim\mathcal{D}}\left[\left\|Q_g(s,a,g_{sub}^h) - r_g\right\|^2\right]. \tag{10}$$

The online low-level policy is updated by simultaneously maximizing $Q$ and $Q_g$:

$$\pi_\theta^l = \arg\max_\theta \mathbb{E}_{(s,g_{sub}^h)\sim\mathcal{D},a\sim\pi_\theta^l(\cdot|s,g_{sub}^h)}[Q(s,a,g_{sub}^h) + \beta \cdot Q_g(s,a,g_{sub}^h)], \tag{11}$$

where $\beta$ is used to control the strength of guidance.

## 5 Experiments

Our experiments delve into the utilization of non-expert observation data to expedite online learning in goal-reaching tasks, particularly focusing on addressing challenges associated with long-horizon objectives. We evaluate the efficacy of our methods on challenging tasks with sparse rewards, as depicted in Figure 2. These tasks encompass manipulation tasks such as SawyerReach [77] and FetchReach [78], as well as long-horizon tasks like FetchPickAndPlace [78], CALVIN [79], and two navigation tasks [80] of varying difficulty levels. Detailed information regarding our evaluation environments can be found in the Appendix. Our experiments aim to provide concrete answers to the following questions:

(1) *Is our method more efficient compared to other online methods and can it compete with offline-online methods learned from fully labeled data?*

(2) *Does the efficiency of our method stem from informative exploration?*

(3) *Do reasonable subgoals make the guidance clearer in our method?*

(4) *Is our method robust to insufficient, low-quality, and highly diverse observation data?*

### 5.1 Comparison with previous methods

**Baselines.** We compare our approach against various online learning methods, including the actor-critic method Online [69], exploration-based methods such as RND [4] and ExPLORe [6], as well

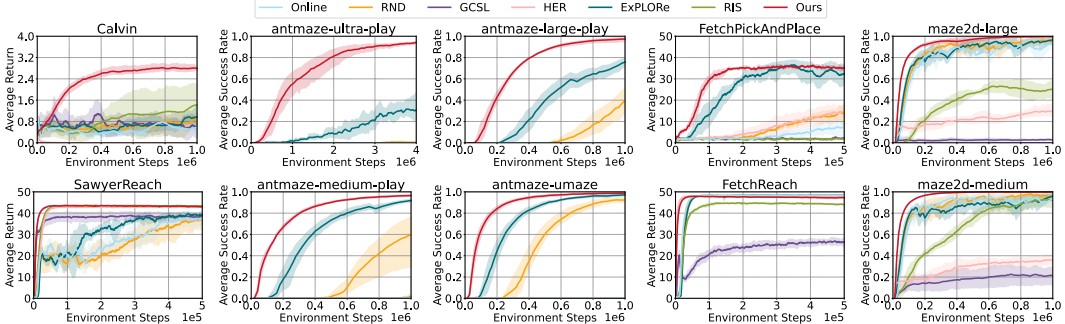

Figure 3: Comparison with online learning methods on robotic manipulation and navigation tasks. Shaded regions denote the 95% confidence intervals across 5 random seeds. Best viewed in color.

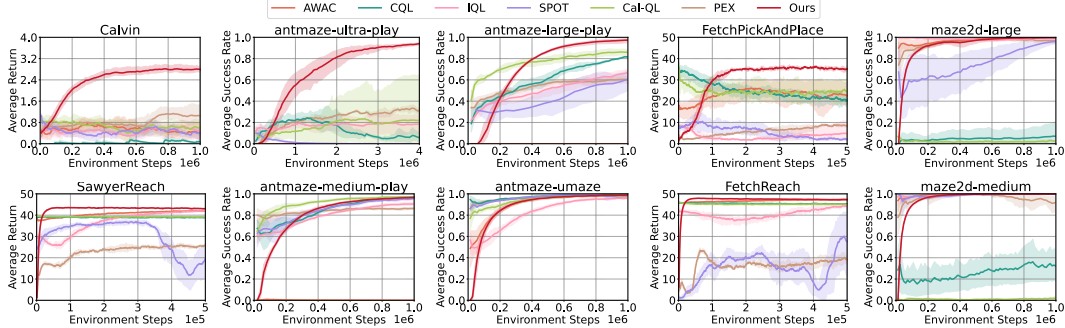

Figure 4: Comparison with offline pre-training and online fine-tuning methods. Shaded regions denote the 95% confidence intervals across 5 random seeds. Best viewed in color.

as data-efficient methods like HER [81], GCSL [82], and RIS [83]. Additionally, we compare against offline-online methods, including naïvely online fine-tuning methods [13] like AWAC [12] and IQL [9], pessimistic methods CQL [8] and Cal-QL [11], policy-constraining method SPOT [24], and policy expansion approach PEX [10]. We set the update data ratio (UTD) [16] to 1 for fair policy updates. Learning curves are presented in Figure 3 and Figure 4.

Figure 3 depicts the performance curves of our approach compared to various online learning methods in navigation and manipulation tasks. Our method shows significant improvements in learning efficiency and policy performance, particularly in challenging tasks with long-horizon challenges such as FetchPickAndPlace, AntMaze-Ultra, and CALVIN. In these tasks, the agent requires intelligent exploration rather than exhaustive exploration of all states, which would be time-consuming. Notably, our approach achieves rapid convergence and surpasses the performance of previous methods while maintaining stability and demonstrating superior efficiency. Figure 4 illustrates a comparison between our approach and offline-online methods. Our method quickly reaches performance levels comparable to prior methods while outperforming them in terms of long-horizon tasks.

## 5.2 Does the efficiency of our method stem from informative exploration?

We evaluated the impact of our method on the state coverage [6] of the agent in the AntMaze domain to investigate whether its effectiveness primarily stems from informative exploration. This evaluation allows us to determine the effectiveness of incorporating non-expert action-free observation data in accelerating online learning. We assess the state coverage achieved by various methods in the AntMaze task, with a specific focus on their exploration effectiveness in navigating the maze. Figure 5 provides a visual illustration of the state visitation on the antmaze-large task. Remarkably, our method guides the agent to explore, prioritizing states that can lead to the final goal.

To provide a more quantitative assessment, we utilize the "weighted state coverage" metric. We divide the map into a grid and assign importance values to each state based on their distance to the goal, as depicted in Figure 6. The weighted coverage metric reflects the average importance of states

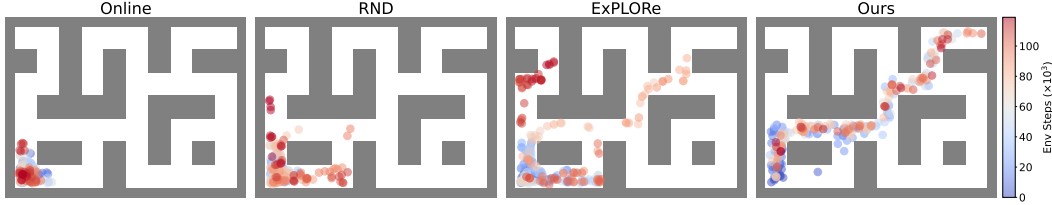

Figure 5: Visualizations of the agent's exploration behaviors on *antmaze-large*. The dots are uniformly sampled from the online replay buffer and colored by the training environment step. The visualization results are obtained by sampling $512$ points from a maximum of $120K$ environment steps. The results show that *Ours* achieves higher learning efficiency via informative explorations.

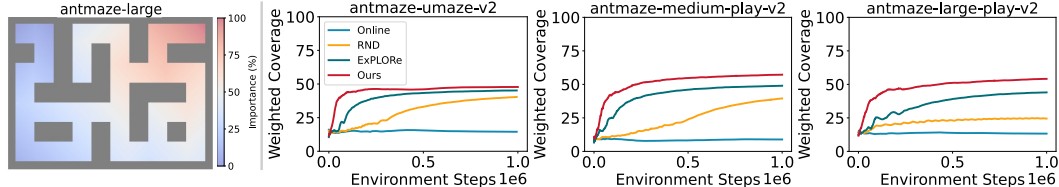

Figure 6: **Left:** Visualization of state importance, which is determined by distance to the goal. **Right:** Weighted coverage for 3 AntMaze tasks. The weighted coverage represents the average importance every $1K$ steps. Higher weighted coverage emphasizes exploration as more valuable. encountered every $1K$ steps. Higher weighted coverage indicates a greater emphasis on exploring important states. In comparison, RND [4] and ExPLORe [6] prioritize extensive exploration but may not necessarily focus on crucial states for task completion. In contrast, our approach concentrates exploration on states more likely to lead to the goal.

### 5.3 Do reasonable subgoals make the guidance more clear?

The learned state-goal value function may erroneously create the perception that it can effectively guarantee informative exploration. However, the learned state-goal value function proves to be imperfect due to the following reasons: (1) Estimating long-horizon goals is likely to introduce a higher degree of noise. (2) As the distance increases, the gradient within the state-goal value function progressively diminishes in prominence. We visualize these errors in the Figure 7, which highlight the limited guidance provided by relying solely on the value function. However, by setting subgoals that offer nearby targets to the current state, we can effectively alleviate the aforementioned errors, especially in the context of long-horizon tasks. In Figure 7(b), the exploration rewards derived from the subgoals demonstrate notable distinctions and enhanced precision, rendering the learning signal more evident and discernible. The ablation study on "w/o subgoals" is presented in the Appendix.

### 5.4 Is our method robust to different prior data?

To further evaluate the capability of our method in leveraging prior data, we modified the *antmaze-large-play-v2* dataset to evaluate our approach under different data corruptions. We primarily consider the following scenarios and report the results in Figure 8.

**Diversity:** We evaluate the sensibility of our method to the diversity of offline trajectories under two variations of *antmaze-large* dataset: *antmaze-large-play*, where the agent navigates from a fixed set of starting points to a fixed set of endpoints, and *antmaze-large-diverse*, where the agent navigates from random starting points to random endpoints.

**Limited Data:** We verify the influence of the quantities of offline trajectories on our performance by removing varying proportions of the data, where $10\%$ denotes only 100 trajectories are preserved.

**Insufficient Coverage:** We assess the dependence of our method on offline data coverage by retaining partial trajectories from the *antmaze-large-play* dataset. We divide it into three regions: *Begin*, *Medium* and *Goal*. We conduct ablation experiments by removing data from each region.

**Incomplete Trajectories:** We verify the robustness of our method to incomplete trajectories by dividing each offline trajectory into segments of varying lengths. We consider three different levels of segmentation: 2 *divide*, 3 *divide* and 4 *divide*. The trajectory lengths are reported in Figure 8(d).

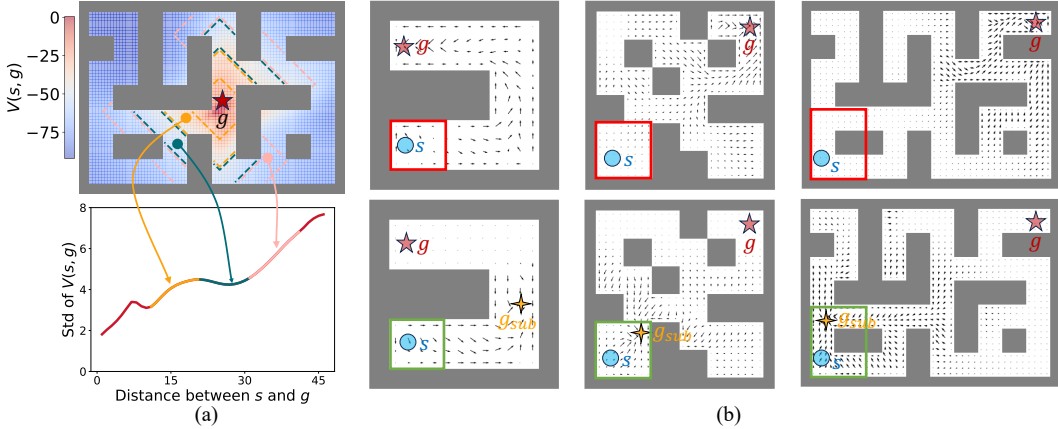

Figure 7: (a) Visualization of the standard deviation of $V(\cdot, g)$: As the distance between the state and the goal increases, the learned value function becomes noisy. (b) **Top:** Relying solely on long-horizon goals leads to unclear and erroneous guidance. **Bottom:** Subgoals make guidance clear. The arrows represent the gradient of $V(\cdot, g)$, reflecting guidance for policy exploration.

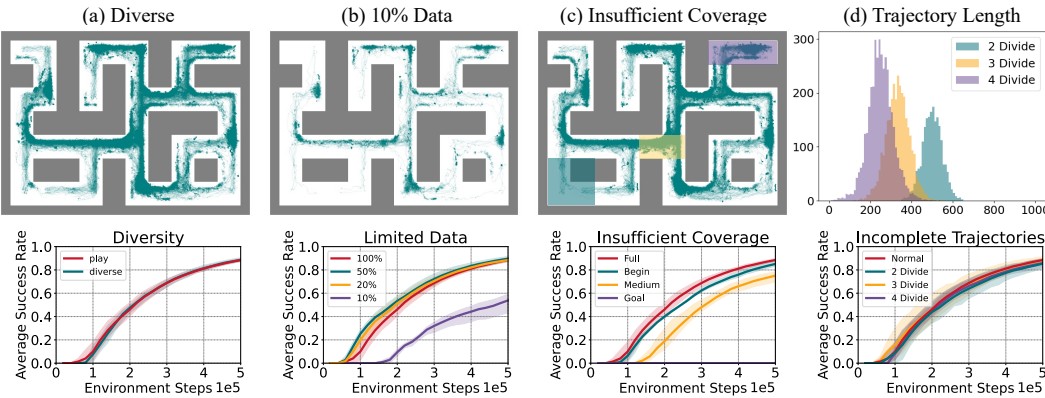

Figure 8: Visualizations of four different types of prior observation data and evaluation results on them. **Top**: Visualization of data characteristics. **Bottom**: Evaluation results. (a) Complexity and Diversity. (b) Limited Dataset Regime. (c) Insufficient Coverage. (d) Incomplete Trajectories.

Our method demonstrates robustness in handling *diverse* datasets, even in scenarios with limited data. It remains effective and stable even when data is extremely scarce. The absence of *Begin* and *Medium* data does not significantly impact the learning of our policy when there is insufficient coverage. However, challenges arise in online learning when there is inadequate coverage of the *Goal* region, highlighting the importance of goal coverage. When dealing with data consisting of trajectory segments, our method excels due to its strong trajectory stitching capability. The above comparison emphasizes the robustness of our method across different datasets, encompassing diverse prior data and low-quality data with varying levels of corruption.

## 6 Conclusion

In conclusion, our proposed method, EGR-PO, addresses the challenging problem of long-horizon goal-reaching policy learning by leveraging non-expert, action-free observation data. Our method learns a high-level policy to generate reasonable subgoals and a state-goal value function to encourage informative exploration. The subgoals, serving as waypoints, provide clear guidance and enhance the accuracy of predicting exploration rewards. These two components naturally integrated into the actor-critic framework, making it straightforward to apply existing algorithms. Our method demonstrates significant improvements over existing goal-reaching methods and shows robustness to various corrupted datasets, enhancing the practicality and applicability of our approach.

## Acknowledgements

This work was supported in part by the National Natural Science Foundation of China under grant U23B2011, 62102069, U20B2063 and 62220106008, the Key R&D Program of Zhejiang under grant 2024SSYS0091, and the New Cornerstone Science Foundation through the XPLORER PRIZE.

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

# A  Implementation Details

## A.1  Diffusion policy

Diffusion probabilistic models [84, 65] are a type of generative model that learns the data distribution $q(x)$ from a dataset $\mathcal{D} := \{x_i\}_{0 \leq i < M}$. It represents the process of generating data as an iterative denoising procedure, denoted by $p_\theta(x_{i-1}|x_i)$ where $i$ is an indicator of the diffusion timestep. The denoising process is the reverse of a forward diffusion process that corrupts input data by gradually adding noise and is typically denoted by $q(x_i|x_{i-1})$. The reverse process can be parameterized as Gaussian under the condition that the forward process obeys the normal distribution and the variance is small enough: $p_\theta(x_{i-1}|x_i) = \mathcal{N}(x_{i-1}|\mu_\theta(x_i, i), \Sigma_i)$, where $\mu_\theta$ and $\Sigma$ are the mean and covariance of the Gaussian distribution, respectively. The parameters $\theta$ of the diffusion model are optimized by minimizing the evidence lower bound of negative log-likelihood of $p_\theta(x_0)$, similar to the techniques used in variational Bayesian methods: $\theta^* = \arg\min_\theta -\mathbb{E}_{x_0}[\log p_\theta(x_0)]$. For model training, a simplified surrogate loss [65] is proposed based on the mean $\mu_\theta$ of $p_\theta(x_{i-1}|x_i)$, where the mean is predicted by minimizing the Euclidean distance between the target noise and the generated noise: $\mathcal{L}_{\text{denoise}}(\theta) = \mathbb{E}_{i,x_0 \sim q, \epsilon \sim \mathcal{N}}[|\epsilon - \epsilon_\theta(x_i, i)|^2]$, where $\epsilon \sim \mathcal{N}(\mathbf{0}, \mathbf{I})$.

Specifically, our diffusion policy is represented as Equation (4) via the reverse process of a conditional diffusion model, but the reverse sampling, which requires iteratively computing $\epsilon_\phi$ networks $N$ times, can become a bottleneck for the running time. To limit $N$ to a relatively small value, with $\beta_{\min} = 0.1$ and $\beta_{\max} = 10.0$, we follow [85] to define:

$$\beta_i = 1 - \alpha_i = 1 - e^{-\beta_{\min}(\frac{1}{N}) - 0.5(\beta_{\max} - \beta_{\min})\frac{2i-1}{N^2}}, \tag{12}$$

which is a noise schedule obtained under the variance preserving SDE of [86].

## A.2  Goal distributions

We train our state-goal value function and high-level policy respectively with Equation (3) and (7), using different goal-sampling distributions. For the state-goal value function (Equation (3)), we sample the goals from either random states, futures states, or the current state with probabilities of 0.3, 0.5, and 0.2, respectively, following [28]. We use $\text{Geom}(1 - \gamma)$ for the future state distribution and the uniform distribution over the offline dataset for sampling random states. For the high-level policy, we mostly follow the sampling strategy of [87]. We first sample a trajectory $(s_0, s_1, \ldots, s_t, \ldots, s_T)$ from the dataset $D_O$ and a state $s_t$ from the trajectory. we either (i) sample $g$ uniformly from the future states $s_{t_g}$ $(t_g > t)$ in the trajectory and set the target subgoal $g_{sub}$ to $s_{\min(t+k, t_g)}$ or (ii) sample $g$ uniformly from the dataset and set the target subgoal to $s_{\min(t+k, T)}$.

## A.3  Advantage estimates

Following [14], the advantage estimates for Equation (6) is given as:

$$\tilde{A}(s_t, s_{t+\tilde{k}}, g) = \gamma^{\tilde{k}} V_\theta(s_{t+\tilde{k}}, g) + \sum_{t'=t}^{\tilde{k}-1} r(s_{t'}, g) - V_\theta(s_t, g), \tag{13}$$

where we use the notations $\tilde{k}$ and $\tilde{s}_{t+k}$ to incorporate the edge cases discussed in the previous paragraph (i.e., $\tilde{k} = \min(k, t_g - t)$ when we sample g from future states, $\tilde{k} = \min(k, T - t)$ when we sample $g$ from random states, and $\tilde{s}_{t+k} = s_{\min(t+k,T)}$). Here, $s_{t'} \neq g$ and $s_t \neq \tilde{s}_{t+k}$ always hold except for those edge cases. Thus, the reward terms in Equation (13) are mostly constants (under our reward function $r(s, g) = 0$ (if $s = g$), $-1$ (otherwise)), as are the third terms (with respect to the policy inputs). As such, we practically ignore these terms for simplicity, and this simplification further enables us to subsume the discount factors in the first terms into the temperature hyperparameter $\beta$. We hence use the following simplified advantage estimates, which we empirically found to lead to almost identical performances in our experiments:

$$\tilde{A}(s, g_{sub}, g) = V_\theta(g_{sub}, g) - V_\theta(s, g), \tag{14}$$

where we use $g_{sub}$ to represent $s_{t+\tilde{k}}$ under various conditions.

Table 1: Hyperparameters.

| Hyperparameter | Value |
|---|---|
| Batch Size | 1024 |
| High-level Policy MLP Dimensions | (256, 256) |
| State-Goal Value MLP Dimensions | (512,512,512) |
| Representation MLP Dimensions | (512,512,512) |
| Nonlinearity | GELU [88] |
| Optimizer | Adam [89] |
| Learning Rate | 0.0003 |
| Target Network Smoothing Coefficient | 0.005 |
| AWR Temperature Parameter | 1.0 |
| IQL Expectile $\tau$ | 0.7 |
| Discount Factor $\gamma$ | 0.99 |
| Diversity of Subgoals $\alpha$ | 0.5 |

## B    Hyperparameters

We present the hyperparameters used in our experiments in Table 1, where we mostly follow the network architectures and hyperparameters used by [28, 14]. We use layer normalization [90] for all MLP layers and we use normalized 10-dimensional output features for the goal encoder of state-goal value function to make them easily predictable by the high-level policy, as discussed in Appendix A.

For the subgoal steps $k$, we use $k = 50$ (AntMaze-Ultra), $k = 15$ (FetchReach, FetchPickAndPlace, and SawyerReach), or $k = 25$ (others). We sample goals for high-level or flat policies from either the future states in the same trajectory (with probability 0.7) or the random states in the dataset (with probability 0.3). During training, we periodically evaluate the performance of the learned policy at every 20 episode using 50 rollouts.

## C    Ablation Study Results

**Subgoal Steps.**  In order to examine the impact of subgoal step values ($k$) on performance, we conduct an evaluation of our method on AntMaze tasks. We employ six distinct values for $k \in \{1, 5, 15, 25, 50, 100\}$. The results, depicted in Figure 9, shed light on the relationship between $k$ and performance outcomes. Remarkably, our method consistently demonstrates superior performance when $k$ falls within the range of 25 to 50, which can be identified as the optimal range. Our method exhibits commendable performance even when $k$ deviates from this range, except in cases where $k$ is excessively small. These findings underscore the resilience and efficacy of our method across various subgoal step values.

**Ablation on Subgoals and Exploration Guidance.**  To demonstrate how subgoals and exploration guidance contribute to efficient policy learning for goal-reaching tasks, we conduct ablation experiments where we remove each component separately. The results, as shown in the Figure 10, highlight the crucial importance of subgoal setting, as the absence of subgoals hinders the resolu-

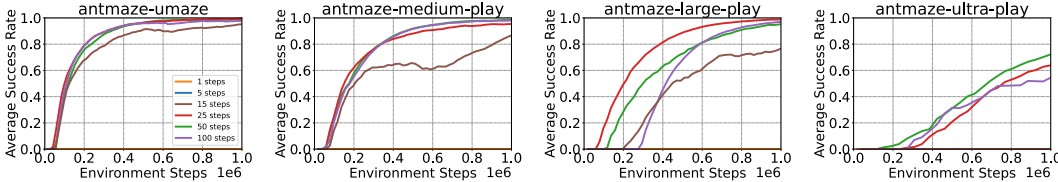

Figure 9: Ablation study of the subgoal steps $k$. Our method generally achieves the best performances when $k$ is between 25 and 50. Even when k is not within this range, ours mostly maintains reasonably good performance unless $k$ is too small (i.e., $\leq 5$).

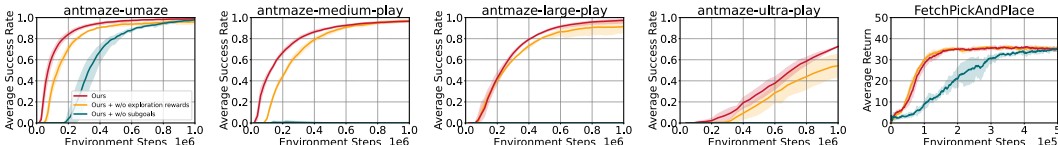

Figure 10: Ablation study on Subgoals and Exploration Guidance. The result shows that the crucial importance of subgoal setting. Additionally, incorporating exploration guidance facilitates the policy in efficiently reaching subgoals, resulting in further improvements in learning efficiency. Shaded regions denote the 95% confidence intervals across 5 random seeds.

tion of long-horizon tasks. Additionally, incorporating exploration guidance facilitates the policy in efficiently reaching subgoals, resulting in further improvements in learning efficiency. Overall, our findings indicate that including both subgoal setting and exploration guidance enables our approach to leverage the benefits of both, leading to efficient learning efficiency.

**Visualization of End-Effector Trajectories in the FetchReach Task.** We fixed the goal at a farther location from the starting point, thereby increasing the task difficulty. In the Figure 11, the **black dot** represents the starting point, and the **green dot** represents the final goal. As shown in the visualization, other methods explore outward evenly from the start point. Despite incorporating active exploration, they struggle to reach the goal point due to the vast state space. Our method, on the other hand, avoids aimless exploration and efficiently reaches the goal point, which aligns with our conclusions.

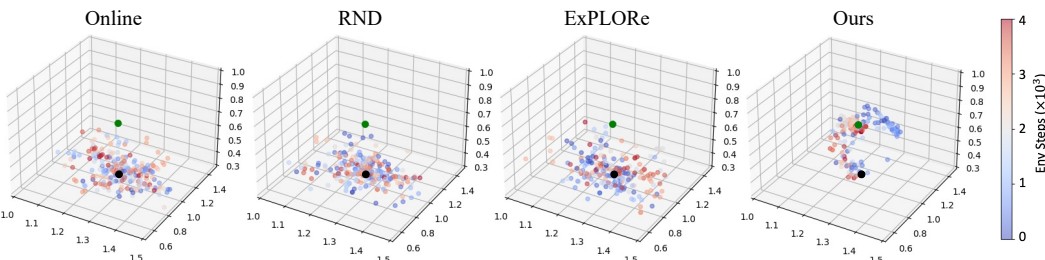

Figure 11: **Visualization of End-Effector Trajectories in the FetchReach Task.** The dots are uniformly sampled from the online replay buffer and colored by the training environment step.

## D   Environments

**SawyerReach** environment, derived from multi-world, involves the Sawyer robot reaching a target position with its end-effector. The observation and goal spaces are both 3-dimensional Cartesian coordinates, representing the positions. The state-to-goal mapping is a simple identity function, $\phi(s) = s$, and the action space is 3-dimensional, determining the next end-effector position.

**FetchReach** and **FetchPickAndPlace** environments in OpenAI Gym feature a 7-DoF robotic arm with a two-finger gripper. In FetchReach, the goal is to touch a specified location, while Fetch-PickAndPlace involves picking up a box and transporting it to a designated spot. The state space comprises 10 dimensions, representing the gripper's position and velocities, while the action space is 4-dimensional, indicating gripper movements and open/close status. Goals are expressed as 3D vectors for target locations.

**Maze2D** is a goal-conditioned planning task, which involves guiding a 2-DoF ball that can be force-actuated in the cartesian directions of x and y. Given the starting location and the target location, the policy is expected to find a feasible trajectory that reaches the target from the starting location avoiding all the obstacles.

**AntMaze** is a class of challenging long-horizon navigation tasks where the objective is to guide an 8-DoF Ant robot from its initial position to a specified goal location. We evaluate the performance

in four different difficulty settings, including the "umaze", "medium" and "large" maze datasets from the original D4RL benchmark. While the large mazes already pose a significant challenge for long-horizon planning, we also introduce an even larger maze "ultra" proposed by [91]. The maze in the AntMaze-Ultra task is twice the size of the largest maze in the original D4RL dataset. Each dataset consists of 999 length-1000 trajectories, in which the Ant agent navigates from an arbitrary start location to another goal location, which does not necessarily correspond to the target evaluation goal. At test time, to specify a goal g for the policy, we set the first two state dimensions (which correspond to the x-y coordinates) to the target goal given by the environment and the remaining proprioceptive state dimensions to those of the first observation in the dataset. At evaluation, the agent gets a reward of 1 when it reaches the goal.

**CALVIN** is another long-horizon manipulation environment features four target subtasks. We use the offline dataset provided by [92], which is based on the teleoperated demonstrations from [79]. The dataset consists of 1204 length-499 trajectories. In each trajectory, the agent achieves some of the 34 subtasks in an arbitrary order, which makes the dataset highly task-agnostic [92]. At test time, to specify a goal g for the policy, we set the proprioceptive state dimensions to those of the first observation in the dataset and the other dimensions to the target configuration. At evaluation, the agent gets a reward of 1 whenever it achieves a subtask.

# E   More Related Work

**Learning Efficiency.** Introducing relabeling can enhance learning efficiency. HER [81] relabels the desired goals in the buffer with achieved goals in the same trajectories. CHER [93] goes a step further by integrating curriculum learning with the curriculum relabeling method, which adaptively selects the relabeled goals from failed experiences. Drawing from the concept that any trajectory represents a successful attempt towards achieving its final state, GCSL [82], inspired by supervised imitation learning, iteratively relabels and imitates its own collected experiences. [94] filters the actions from demonstrations by Q values and adds a supervised auxiliary loss to the RL objective to improve learning efficiency. RIS [83] uses imagined subgoals to guide the policy search process. However, such methods are only useful if the data distribution is diverse enough to cover the space of desired behaviors and goals and may still face challenges in hard exploration environments.

# F   Baseline Introduction

## F.1   Online learning baselines

**Online:** A standard off-policy actor-critic algorithm [69] which trains an actor network and a critic network simultaneously from scratch that does *not* make use of the prior data at all.

**RND:** Extends the *Online* method by incorporating Random Network Distillation [4] as a novelty bonus for exploration. given an online transition $(s, a, r, s')$, and RND feature networks $f_\phi(s, a)$, $\bar{f}(s, a)$, we set

$$\hat{r}(s, a) \leftarrow r + \frac{1}{L}||f_\phi(s, a) - \bar{f}(s, a)||_2^2 \qquad (15)$$

and use the transition $(s, a, \hat{r}, s')$ in the online update. The RND training is done the same way as in our method where a gradient step is taken on every new transition collected.

**HER:** Combines *Online* method with Hindsight Experience Replay [81] to improve data efficiency by re-labeling past data with different goals.

**GCSL:** Trains the policy using supervised learning, leading to stable learning progress.

**RIS:** This method [83] incorporates a separate high-level policy that predicts intermediate states halfway to the goal. By aligning the subgoal reaching policy with the final policy, RIS effectively regularizes the learning process and improves performance in complex tasks.

**ExPLORe:** This approach learns a reward model from online experience, labels the unlabeled prior data [6] with optimistic rewards, and then uses it concurrently alongside the online data for downstream policy and critic optimization.

### F.2 offline-online baselines

**AWAC:** AWAC combines sample-efficient dynamic programming with maximum likelihood policy updates, providing a simple and effective framework that is able to leverage large amounts of offline data and then quickly perform online fine-tuning of reinforcement learning policies.

**IQL:** Avoiding querying out-of-sample actions by converting the max operator in the Bellman optimal equation into expectile regression,and thus learn a better Q Estimation.

**CQL:** CQL imposes an additional regularizer that penalizes the learned Q-function on out-of-distribution (OOD) actions while compensating for this pessimism on actions seen within the training dataset. Assuming that the value function is represented by a function, $Q_\theta$ , the training objective of CQL is given by

$$\min_{\boldsymbol{\theta}} \alpha \underbrace{\left(\mathbb{E}_{s\sim\mathcal{D},a\sim\pi}\left[Q_\theta(s,a)\right] - \mathbb{E}_{s,a\sim\mathcal{D}}\left[Q_\theta(s,a)\right]\right)}_{\text{Conservative regularizer } \mathcal{R}(\theta)} + \frac{1}{2}\mathbb{E}_{s,a,s'\sim\mathcal{D}}\left[\left(Q_\theta(s,a) - \mathcal{B}^\pi\bar{Q}(s,a)\right)^2\right],$$

(16)

where $\mathcal{B}^\pi\bar{Q}(s,a)$ is the backup operator applied to a delayed target Q-network, $\bar{Q}$: $\mathcal{B}^\pi\bar{Q}(s,a) := r(s,a) + \gamma E_{a'\sim\pi(a'|s')}[\bar{Q}(s',a')]$. The second term is the standard TD error. The first term $R(\theta)$ is a conservative regularizer that aims to prevent overestimation in the Q-values for OOD actions by minimizing the Q-values under the policy $\pi(a|s)$, and counterbalances by maximizing the Q-values of the actions in the dataset following the behavior policy $\pi_\beta$.

**Cal-QL:** This method learns a conservative value function initialization can speed up online fine-tuning and harness the benefits of offline data by underestimating learned policy values while ensuring calibration. Specifically, Calibrating CQL constrain the learned Q-function $Q_\theta^\pi$ to be larger than value function $V$ via a simple change to the CQL training objective. Cal-QL modifies the CQL regularizer, $R(\theta)$ in this manner:

$$\mathbb{E}_{s\sim\mathcal{D},a\sim\pi}\left[\max\left(Q_\theta(s,a), V(s)\right)\right] - \mathbb{E}_{s,a\sim\mathcal{D}}\left[Q_\theta(s,a)\right], \tag{17}$$

where the changes from standard CQL are depicted in red.

**SPOT:** This work constrains the policy network in offline reinforcement learning (RL) to not only be within the support set but also avoid the out-of-distribution actions effectively unlike the standard behavior policy through behavior regularization.

**PEX:** This work introduces a policy expansion scheme. After learning the offline policy, it is included as a candidate policy in the policy set, which further assists in learning the online policy. This method avoids fine-tuning the offline policy, which could disrupt the learned policies, and instead allows the offline policy to participate in online exploration adaptively.

