# OpenReview forum: "Goal-Reaching Policy Learning from Non-Expert Observations via Effective Subgoal Guidance"
_robot-learning.org/CoRL/2024/Conference — CoRL 2024_

### Official Review · Reviewer_RXHf · 2024-07-21

**Originality:** 3
**Technical Quality:** 3
**Clarity Of Presentation:** 4
**Potential Impact:** 2
**Recommendation:** 3
**Confidence:** 4

**Review:**

Strengths
- In general, the quality of paper is good with a clear writing. Experiments are thorough and supporting the claims made in the paper.
- The paper is dealing with the practical and important setup of utilizing action-free prior data for accelerating reinforcement learning.

Weaknesses
- The paper is missing experiments in a setup more relevant to real-world robotic setup, for instance, all the experiments are conducted in a state-based benchmarks without vision inputs. Can this method be scalable to setups with visual observations?
- One major limitation I think is that high-level policy is fixed after training with prior observations. While the proposed idea can be robust to some scenerios where prior data is not perfect in Section 5.4, not updating the high-level policy can severely limit the asymptotic performance of the proposed idea, especially when the prior data consists of small amounts of initial exploratory data.
- Can analysis similar to Figure 5 be conducted for manipulation tasks where the gain from the current idea is significant? For instance, what would the end-effector trajectories or joint positions look like throughout the training?
- Minor: move a part that describes how goals are sampled (line 135-137) to maybe line 128, as there's no description on what g means for Equation (3)

**Quality Of The Limitations Section:**

1

**Questions For Rebuttal:**

- Please address my weaknesses in the main review.

**Robotics Focus:**

2

**Summary Of Paper:**

This paper introduces a goal-conditioned RL framework that can utilize prior data for guiding the learning procedure. Specifically this paper (i) learns a state-goal value functions from prior action-free data and use it for informative exploration and (ii) learns a high-level policy with diffusion models that can generate nearby subgoals. Proposed components are integrated into actor-critic algorithm. Experiments are conducted on a variety of state-based benchmarks.

**Summary Of Recommendation:**

This paper is of good quality but its relevance to real-world robotics is not high. I'm in a borderline between weak accept / weak reject evaluations for this paper, I'd like to read the reviews of other reviewers and read the rebuttal response from the authors before making a final recommendation decision.

---

### Official Review · Reviewer_qsNz · 2024-07-22
**Goal-Reaching Policy Learning from Non-Expert  Observations via Effective Subgoal Guidance**

**Originality:** 3
**Technical Quality:** 3
**Clarity Of Presentation:** 2
**Potential Impact:** 2
**Recommendation:** 3
**Confidence:** 3

**Review:**

This paper proposes a framework that combines subgoal generation and state-goal value function learning. The quality of the writing is good, and its motivation is presented well.
Clarity: The framework is clear. However, some details need to be provided. For instance, the reward function of each environment is not provided. If each the reward is a sparse reward, the success of each environment needs to be defined.
Originality: There are some similar works, but the proposed approach is original.
Significance: The main idea is not very far from HER, and many similar works. The task seems very simple if the observations are all the low dimensional states.
Strengths:
The paper is well structured.
The article conducted a lot of comparative experiments and verified the results in multiple environments.
Weaknesses：
Some details about the task are missing.
The way of generation of prior data is not provided.
The ablation study is weak. The offline methods (like HER) directly use the prior data need to be included.

**Quality Of The Limitations Section:**

2

**Questions For Rebuttal:**

Please refer to the weaknesses.

**Robotics Focus:**

2

**Summary Of Paper:**

This paper proposed a hierarchical policy learning strategy combining subgoal generation and state-goal value function learning.

**Summary Of Recommendation:**

Overall, the method is efficient with sufficient experiments provided. Although I worried about its significance in real world, I recommend accepting the paper, but will not argue for my recommendation if other reviewers have a different opinion.

---

### Official Review · Reviewer_onEe · 2024-07-28

**Originality:** 2
**Technical Quality:** 3
**Clarity Of Presentation:** 3
**Potential Impact:** 2
**Recommendation:** 3
**Confidence:** 4

**Review:**

1. Subgoal generation

Suggoals are generated using a combination of the action-free state-goal value function, and advantage weighted imitation of states that either appear in the future of the trajectory, or are randomly picked from the dataset. Why are randomly sampled goals reasonable subgoals? These could be arbitrarily far away from the current state, making them infeasible. Further, if the trajectory is sub-optimal, then picking future states in the trajectory will not necessarily be reasonable. If the action-free state-goal value function is unreliable with increasing distance, then it in unclear why resorting to states that are potentially arbitrarily far away or from suboptimal trajectories is the right correction. There is prior work that trains world models on potentially suboptimal prior data [1,2], enabling accurate long horizon prediction and showing efficient exploration. These works have not been mentioned or compared against.


2. Exploration reward

Exploration is induced in goal conditioned policy learning using a specific reward defined in Equation 9, which compares the state-goal value function at the current and next states. It is not immediately clear why this formulation should promote exploratory behavior, and is not discussed in the paper. If the motivation relates to a notion of uncertainty, the authors should compare to prior work on exploration using uncertainty estimation, via prediction error [3], model disagreement [4], and world model disagreement for long horizon prediction [1,2]. This comparison is important since the exploration ability of the method is stressed as an important point, and this is the key part of the algorithm that encourages exploration.

[1] : Planning to Explore via Self-Supervised World Models, Sekar et al.
[2] : Discovering and Achieving Goals via World Models, Mendonca et al.
[3] : Curiosity-driven Exploration by Self-supervised Prediction, Pathak et al.
[4] : Self-Supervised Exploration via Disagreement, Pathak et al.

**Quality Of The Limitations Section:**

1

**Questions For Rebuttal:**

1. Please address the questions regarding subgoal generation listed above.
2. Please address questions regarding exploration reward above, add comparisons requested.

**Robotics Focus:**

2

**Summary Of Paper:**

Proposes an approach from learning from sub-optimal data by learning to propose reasonable subgoals. This is used with a learned goal-conditioned policy. Experiments are conducted on some mujoco environments involving arms and mazes for locomotion.

**Summary Of Recommendation:**

Due to the concerns regarding subgoal generation, the exploration reward, and missing comparisons, I vote for rejection.

---

### Author Rebuttal · Authors · 2024-08-11

We sincerely appreciate the valuable feedback provided by the reviewers (onEe, qsNz, and RXHf). We have carefully reviewed and addressed each concern raised in their comments. **The detailed responses can be found in the attached .zip file.**

Thank you once again for your constructive insights, which have significantly contributed to improving our work.

---

### Decision · Program_Chairs · 2024-09-04

**Decision:**

Accept

**Comment:**

Initially the reviewers raised concerns regarding the exploration rewards used and random subgoal sampling, among others. The rebuttal submitted by the authors addressed these concerns to a large extent and now all reviewers uniformly recommend acceptance.